# Stereotactic Body Radiotherapy versus Surgery for Lung Metastases from Colorectal Cancer: Single-Institution Results

**DOI:** 10.3390/cancers15041195

**Published:** 2023-02-13

**Authors:** Nagore Garcia-Exposito, Ricard Ramos, Valentin Navarro-Perez, Kevin Molina, Maria Dolores Arnaiz, Susana Padrones, Jose Carlos Ruffinelli, Cristina Santos, Ferran Guedea, Arturo Navarro-Martin

**Affiliations:** 1Radiation Oncology Department, Institut Català d’Oncologia, 08908 Barcelona, Spain; 2Department of Thoracic Surgery, Hospital Universitari de Bellvitge, Bellvitge Biomedical Research Institute (IDI-BELL), L’Hospitalet de Llobregat, 08908 Barcelona, Spain; 3Clinical Research Unit, Institut Català d’Oncologia, Bellvitge Biomedical Research Institute (IDIBELL), L’Hospitalet de Llobregat, 08908 Barcelona, Spain; 4Medical Oncology Department, Institut Català d’Oncologia, L’Hospitalet de Llobregat, 08908 Barcelona, Spain; 5Department of Respiratory Medicine, Hospital Universitari de Bellvitge, Bellvitge Biomedical Research Institute (IDIBELL), University of Barcelona, L’Hospitalet de Llobregat, 08908 Barcelona, Spain

**Keywords:** colorectal cancer, lung metastases, oligometastases, stereotactic body radiotherapy, metastasectomy, wedge resection

## Abstract

**Simple Summary:**

Colorectal cancer (CRC) is one of the most common malignancies in developed countries and 50% of patients will progress to metastatic disease during follow-up, the liver and lungs being the most common sites. For lung metastases in particular, although surgery has historically been the treatment of choice, the use of stereotactic body radiotherapy (SBRT) is increasing. SBRT is known to be a well-tolerated and less invasive alternative to surgery, with excellent results in terms of local control and toxicity. The aim of this retrospective, single-center study was to compare local thoracic control rates with SBRT against those with surgery.

**Abstract:**

Background: Surgery and stereotactic body radiotherapy (SBRT) are two of the options available as local treatments for pulmonary oligometastases from colorectal cancer (CRC). We hypothesized that SBRT would have, at least, a similar local control rate to surgery. Methods: We identified an initial cohort of 100 patients with CRC who received SBRT or surgery for lung metastases. This was then narrowed down to 75 patients: those who underwent surgery (n = 50) or SBRT (n = 25) as their first local thoracic treatment between 1 January 2004 and 29 December 2017. The Kaplan–Meier method was used to calculate lung-progression-free survival (L-PFS) and overall survival (OS). Results: The 1 and 2-year L-PFS was 85% and 70% in the surgical group and 87% and 71% in the SBRT group, respectively (*p* = 0.809). No significant differences were found between the two groups in terms of OS. The biologically effective dose (BED), age and initial CRC stage did not have a significant effect on local control or survival. No grade 3 or above acute- or late-toxicity events were reported. Conclusions: These results add retrospective evidence that SBRT and surgery have similar results in terms of OS and local control in patients with lung oligometastases from CRC.

## 1. Introduction

Colorectal cancer (CRC) is described as the third most common malignancy in developed countries. Approximately 20–25% of patients have metastases at initial presentation and 50% will progress to metastatic disease during follow-up, the liver and lungs being the most common sites [1]. Recently, the distinction between oligoprogression and multiprogression has been made, as the former, with a limited number of metastases, is considered amenable to local salvage treatment [2].

For lung metastases in particular, although surgery (metastasectomy or wedge resection) has historically been the treatment of choice [3,4,5], other strategies such as stereotactic body radiotherapy (SBRT) or radiofrequency ablation (RFA) are now also used in this setting [6,7,8], and due to the absence of prospective trials, most knowledge is obtained from retrospective studies.

SBRT is a well-tolerated and less invasive alternative to surgery for patients who decline or are considered unsuitable for lumpectomy [9]. The excellent results in terms of local control and toxicity in patients with primary non-small-cell lung cancer (NSCLC) have now been extrapolated to patients with oligometastatic disease from other primary malignancies [10,11]. Although the toxicity profile is low, the indication of SBRT in ultracentral lesions (hilar or less than 2 cm from airways) is limited due to the risk of high-grade toxicity. Regarding the toxicity risk, published series report local control rates between 65 and 80% at 2 years in patients with lung metastases from CRC treated with SBRT [12,13]. Studies suggest that a high biologically effective dose (BED) of >120 Gy and small treatment volume are the features most associated with better local control [14,15].

The aim of this retrospective, single-center study was to compare local thoracic control rates in patients treated with SBRT vs. surgery; we hypothesized that these would be similar.

## 2. Materials and Methods

### 2.1. Patients

We identified a retrospective cohort of patients from the multidisciplinary lung tumor board database with primary CRC who underwent SBRT or metastasectomy for the treatment of lung metastases at any point during the course of their disease. We termed this lung progression (LP).

Progression was defined as oligoprogression or multiprogression depending on the number of metastases (5 or fewer vs. more than 5, respectively [2]) and the possibility of receiving salvage treatment.

The initial cohort was then narrowed down to those whose first progression was limited to the lungs and who received local thoracic treatment. This was termed first lung progression (FLP) (Figure 1).

Informed consent was obtained prior to treatment (either surgery or radiotherapy). This study was approved by our Institutional Review Board.

### 2.2. Treatment

All patients were discussed in a multidisciplinary meeting that involved thoracic surgeons, medical oncologists, radiation oncologists, radiologists and nuclear medicine physicians.

As previously explained, surgery was the first-choice treatment option. SBRT was proposed when patients were unsuitable for surgery, had previous thoracic interventions or due to patient choice.

The SBRT was delivered in an outpatient setting, using fractionation schemes according to the discretion of the radiation oncologist, and a dose–risk schedule based on lesion size and distance from critical structures.

For treatment simulation and on the day of treatment, patients were immobilized using a thermoplastic mask. During the scan, we used an acoustic signal to notify patients when to breath in and out in order to ensure that the respiratory pattern was stable, in accordance with the Varian Real-Time Position Management System (Varian Medical Systems, Palo Alto, CA, USA). A 3-mm-thickness planning computed tomography (CT) of the chest without contrast was obtained using 4-dimensional (4D) CT imaging to complete a respiratory cycle with each slide.

The gross tumor volume (GTV) was delineated in each 4D-CT phase and identified in the lung parenchyma window; the clinical target volume (CTV) was coincident to the GTV. The internal target volume (ITV) was formed by the envelope of the CTV from each respiratory phase. The planning target volume (PTV) was defined as the ITV with an isotropic margin of 5 mm. Positron-emission tomography (PET)–CT fusion imaging was performed when required to help locate metastases.

SBRT was delivered shaped with multileaf collimators and using multiple fixed coplanar beams or conformal arcs. The prescription was designed to cover the 85% isodose line (permitted between 60 and 90%). When the dose was >105%, it was required to be within the PTV, and when the dose was outside the PTV, a volume < 15% of the PTV was permitted. The target conformity index was <1.2 (a range of 1.2–1.4 was permitted).

RTOG-0236 recommendations were used as a basis for constraints for organs at risk (OAR). Cone-beam CT (CBCT) was performed before each fraction to verify and correct setup errors.

Surgical treatment was based on the clinical judgement of the surgical team and details were not recorded in the present study.

### 2.3. Follow-Up

Follow-up was performed every three months during the first year and once every six months thereafter. Physical examination and blood tests in order to asses carcinoembryonic antigen (CEA) levels were performed during these reviews. Thoracic CT or FDG-PET-CT were also performed as follow-up investigations if considered necessary.

### 2.4. Study Endpoints

#### 2.4.1. Primary Endpoint

The primary endpoint was the time to lung progression from first pulmonary treatment (L-PFS).

Due to the fact that some surgeries were lobectomies (rather than wedge resections or metastasectomies), the possibility of local recurrence in the same lobe did not exist in these patients. In addition, occasionally, progression after surgery or SBRT was seen in the same lobe but not in the previously treated target area. Taking these points into account, we defined lung progression as the occurrence of new metastases in both lungs.

#### 2.4.2. Secondary Endpoints

Overall survival was analyzed for both cohorts (OS-LP for LP cohort and OS-FLP for FLP cohort). The correlation of clinical factors and dosimetry data with survival rates was also analyzed.

The biologically effective dose (BED_10_) was calculated using the linear–quadratic equation and an α/β of 10.

### 2.5. Toxicity

Patients’ medical records were reviewed to obtain information on toxicity. Common Terminology Criteria for Adverse Events (CTCAE) version 4.0 was used for grading and only grade 3 or above were included [16].

### 2.6. Statistics

A retrospective analysis was performed in May 2021.

Lung-progression-free survival (L-PFS) was defined as the interval between the end of the first local lung treatment and the date of diagnosis of a second thoracic tumor metastasis.

For patients who did not undergo a local thoracic treatment for their first progression, overall survival (OS-LP) was defined as the time from first progression to death or last follow-up.

For the FLP cohort, overall survival (OS-FLP) was defined as the period between the date of lung surgery or SBRT and the date of last follow-up or death from any cause (whichever occurred first).

The following parameters were evaluated to determine their influence on survival: treatment modality (SBRT vs. surgery), age (> or ≤ 70 years old), BED (> or ≤ 120 Gy) and initial CRC stage.

Chi-squared or Fisher’s exact tests were used to compare categorical variables between the two treatment groups. The Mann–Whitney U test was used to compare continuous variables. Univariate Cox proportional hazards analyses were performed to evaluate the effect of variables on PFS and OS.

The Kaplan–Meier test was used to estimate the OS and the PFS rates, and differences in survival outcomes were assessed with the log-rank test. Further analysis was limited due to the relatively small sample size and limited number of events.

*p* values < 0.05 were considered significant.

All statistical analyses were performed using SPSS version 26 (IBM Corp., Armonk, NY, USA).

## 3. Results

### 3.1. Patient and Treatment Characteristics

#### 3.1.1. LP Cohort

An initial cohort of 100 patients with lung metastases from CRC treated with either SBRT or metastasectomy between 1 January 2004 and 4 March 2020 was identified.

The first progression was defined as multiple in nine patients and oligo in 91 patients. Of these 91 patients, 75 had lung oligoprogression; the remaining 16 had progression elsewhere (bone, liver, peritoneum, lymph nodes).

#### 3.1.2. FLP Cohort

This cohort comprised the 75 patients whose first progression was lung oligoprogression and who were treated with local salvage therapy between 1 January 2004 and 29 December 2017. Of these 75 patients, 66 presented metachronous lung progression whilst nine patients were classified as synchronous.

Fifty patients underwent metastasectomy as the primary treatment of lung metastasis and 25 patients received SBRT. Of the 75 patients in this cohort, 19 received both local and systemic treatment—15 in the surgery group and 4 in the SBRT group.

Although the two groups were unbalanced in terms of case numbers, their baseline characteristics were generally similar. There was a slight trend towards more elderly patients in the SBRT group (median 75 years in both groups; 44% vs. 56% were > 70 years old in the surgery and SBRT groups, respectively).

As previously mentioned, surgery was considered the treatment of choice. SBRT was performed in the following situations: due to comorbidity (n = 2), patient’s wish (n = 4), previous surgery (n = 1), unresectable (n = 2), not specified (n = 16).

The median BED_10_ for all FLP patients was 122.25 Gy. The most-used SBRT schedules were the following: 50 Gy in 4 fractions (112.5 BED_10_, 8%), 54 Gy in 3 fractions (151.2 BED_10_, 5.3%), 60 Gy in 8 fractions (105 BED_10_, 5.3%), 60 Gy in 5 fractions (132 BED_10_, 2.7%), 50 Gy in 10 fractions (75 BED_10_, 4%), and a monofraction of 34 Gy (149.6 BED_10_, 8%).

Patient and treatment characteristics are displayed in Table 1 and Table 2.

### 3.2. Treatment Outcomes and Follow-Up

#### 3.2.1. Lung Progression Cohort

Median OS was 86 months (range 51.93–78.69 months).

In the initial cohort of 100 patients with lung progression, overall survival (OS-LP) was significantly different between patients with multiprogression and those with oligoprogression (median survival of 24.48 and 71.23 months, respectively; log-rank test, *p* < 0.001, 95% CI 50.35–92.11 months) (Figure 2).

Statistically significant differences in survival were also present between both treatment groups (surgery and SBRT) and the rest of the patients from the LP cohort (those whose first progression was neither oligoprogression nor pulmonary progression) (median survival of 87.82 months in the surgery group, 56.68 in the SBRT group and 28.29 months for the rest of the patients from the LP cohort; *p* = 0.001).

#### 3.2.2. FLP Cohort

Patients were followed up for a median of 36.01 months (range: 0.23–117.52) after the first thoracic intervention.

Univariate analysis showed that age, sex, tumor size and treatment modality did not have a significant influence on PFS.

There were no statistically significant differences between the treatment groups in terms of L-PFS. Median L-PFS was not reached and the 1- and 2-year L-PFS rates were 85% and 70% in the metastasectomy group and 87% and 71% in the SBRT group, respectively (Figure 3).

No significant differences were found in L-PFS on the basis of age, initial CRC stage or treatment modality (Figure A1, Figure A2 and Figure A3).

Twenty-six patients had progression following the first local approach: 20 from surgery (1 multiple, 19 oligoprogression) and 6 from SBRT (all oligoprogression) (*p* = 0.344).

To assess the impact of the total administered dose, a cutoff of BED_10_ > or ≤ 120 Gy was established. There were no differences in terms of local control or OS (Figure A4).

Initial diagnosis stages I and II were grouped because of the small number of events identified in each group. A trend towards better OS was observed in this group of patients, although it was not statistically significant.

There were no statistically significant differences in terms of survival, with 1- and 2-year OS-FLP rates of 96% and 90% with surgery vs. 90% and 89% with SBRT, respectively (*p* = 0.63) (Figure A5).

At the time of the last follow-up, 34% vs. 20% of patients were reported as deceased in the surgery and SBRT groups, respectively. Cause of death was cancer progression in 100% of deaths in the surgery group and 33% of deaths in the SBRT group.

### 3.3. Toxicity

No patients had grade 3 or above toxicity.

## 4. Discussion

This retrospective analysis is, to our knowledge, one of the first comparative studies carried out to compare salvage treatments for lung metastases from primary CRC. The finding of no statistically significant differences between SBRT and surgery in terms of L-PFS is evidence that these two treatment options are equally effective. Moreover, the fact that no patient developed grade 3 or above toxicity supports the use of SBRT as a feasible treatment option.

Clinical indications for local therapies in the metastatic population, as well as the most appropriate treatment approach, remain the subject of debate. Recent studies have highlighted the lack of randomized prospective evidence on the role of surgery in this setting [4,17]. Furthermore, the SBRT cohorts are usually smaller, and patients have a poorer performance status and tend to be older than the surgical cohorts. This can lead to additional difficulties when designing a prospective study and secondary systemic bias when comparing both salvage treatment modalities.

We selected a homogenous population, choosing patients whose first progression was described as pulmonary oligoprogression and who were treated with SBRT or surgery (25 vs. 50 patients, respectively). Although these two groups were unbalanced in terms of case numbers, a stratified analysis was performed according to baseline clinical characteristics (Table 2).

Regarding the radiation dose, SBRT fractionation was extrapolated from treatments used in primary lung malignancies given the excellent results seen in these patients in terms of toxicity and local control (range 75–90% depending on the series) [7,12,18]. Studies suggest that not only the total dose but also the BED is relevant in these patients. A minimum threshold of BED_10_ = 100 Gy has been well established [19], and BED_10_ > 120 Gy is described in the literature with better local control rates [14]. Our analysis did not find differences between these two groups, probably due to the small size of the cohort and the follow-up time.

Since the term “abscopal effect” was first described in 1953 in reference to the effect of radiation at a distance from the irradiation field, countless studies have been developed and many hypotheses proposed [20]. Recent evidence shows that ionizing radiation may enhance immune responses to tumor cells, especially when hypofractionation or other modifications of standard fractionations are used [21]. Moreover, it has been described that T cells mediate distant tumor inhibition caused by radiotherapy [22]. These findings are of great relevance in the context of metastases, and the study of the molecular mechanisms of the ionizing radiation-induced immune response is becoming a topic of great importance. In the present study, we observed that 100% of the deaths in the surgery cohort were from cancer progression, compared with 33% in the SBRT group; this might suggest that the population treated with SBRT is more fragile but also supports the abscopal theory.

The results of our study regarding survival and local control were similar to those described in the literature when comparing SBRT with metastasectomy, with 1- and 2-year L-PFS rates of 87% and 71% vs. 85% and 70%, in the two groups, respectively. Filippi et al. observed a higher rate of recurrence among the SBRT cohort, acknowledging the difficulties in distinguishing such differences from the effects of sample sizes and follow-up protocols. Consequently, their Kaplan–Meier and Cox models described a worse prognosis in terms of PFS, with 1-year PFS rates of 55% with SBRT and 80% with surgery (*p* < 0.001) [23]. Published reports on local control with SBRT for lung metastases from CRC are summarized in Table 3.

In the present study, the overall survival at 2 years—90% in the surgery group vs. 89% in the SBRT group—was superior to a previous series [23], probably due to the fact that the selection criteria were narrowed down. Nelson et al. retrospectively reviewed patients treated with SBRT or wedge resection and described a higher risk of local recurrence after radiotherapy rather than surgery (2-year risk of local recurrence 29.4% vs. 14.1%, respectively). It was suggested that dose escalation may be required for colorectal histology in particular, as evidence of radioresistance has emerged [24].

However, a systematic review of SBRT vs. surgery for patients with pulmonary oligometastases concluded that there were no substantial differences between the two techniques in terms of short-term survival outcomes, but surgery seems to provide better long-term survival [25].

**Table 3 cancers-15-01195-t003:** Published reports of SBRT in patients with lung metastases from CRC.

Study (Year)	Number of Lesions	Median BED_10_	LC rates	Median Follow-Up (Months)	Ref
Qiu et al. (2015)	65	ns Total dose: 40–60 Gy in 5–11 fr	1-year 56.6%	6.4	[7]
Carvajal et al. (2015)	13	149.6 Gy	1- and 2-year 92.3%	9.16	[14]
Jung et al. (2015)	79	ns Total dose: 40–60 Gy in 3–4 fr	1-year 88.7% 3-year 70.6%	42.8	[26]
		≤48 Gy 60 Gy	3-year 64.6% 3-year 84%		
Wegner et al. (2018)	22	Range 60–105.6 Gy	1-year 75% 2-year 65%	28.5	[12]
		<100 Gy ≥100 Gy	1-year 50% 1-year 87%		
Li et al. (2019)	105	100 Gy		14	[15]
		<100 Gy ≥100 Gy	1-year 96% 1-year 80%		
Nicosia et al. (2020)	107	105 Gy	1-year 91.5% 2-year 80%	28	[13]
Present study	25	122.25 Gy	1-year 87% 2-year 71%	36.01	

LC, local control; BED_10_, biologically effective dose; ns, not specified; Gy, gray.

The effect of advances in systemic therapies was not evaluated; further analysis is ongoing as this may have a significant impact. Taking into account the constant investigation of systemic targeted therapies and the possibility of having different molecular profiles during the evolution of the oncological disease, obtaining new histology samples might be beneficial. In this setting, surgery could offer an advantage over SBRT.

A major limitation of this analysis is the small sample size, which might have reduced the capacity to detect meaningful differences between groups. Moreover, we did not specify whether the pulmonary progression occurred in the field of the SBRT treatment or the metastasectomy bed. In this setting, cancer-specific survival (CSS) might be a more representative endpoint. Surgical complications were not registered, and this missing information might also be a limitation when comparing the outcomes of both techniques. However, our study populations were homogeneous, and the criteria for surgery or SBRT were the same.

## 5. Conclusions

This retrospective series suggests that SBRT had comparable outcomes to surgery in terms of L-PFS and OS and represents an alternative option to surgery for the treatment of colorectal lung metastases. Thus, in selected patients, not suitable for surgery and in which no pathological confirmation is needed, it may represent a valuable therapeutic option. However, our findings warrant the need for a prospective trial comparing the surgical approach and SBRT in this specific population.

## Figures and Tables

**Figure 1 cancers-15-01195-f001:**
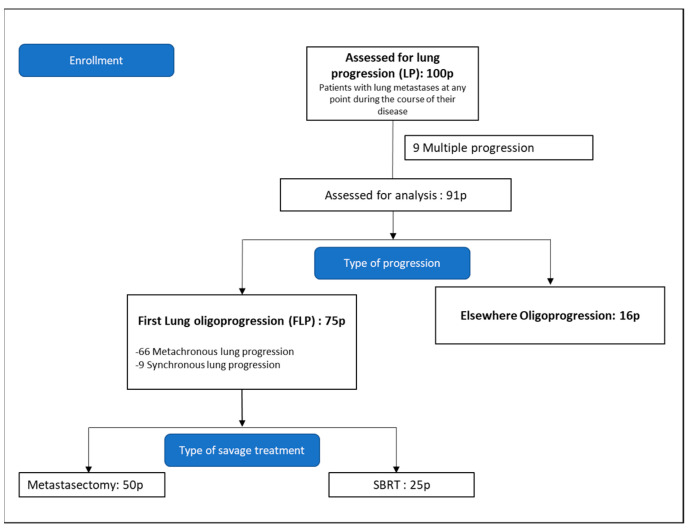
Consort diagram. Pathological confirmation of pulmonary metastases was not required.

**Figure 2 cancers-15-01195-f002:**
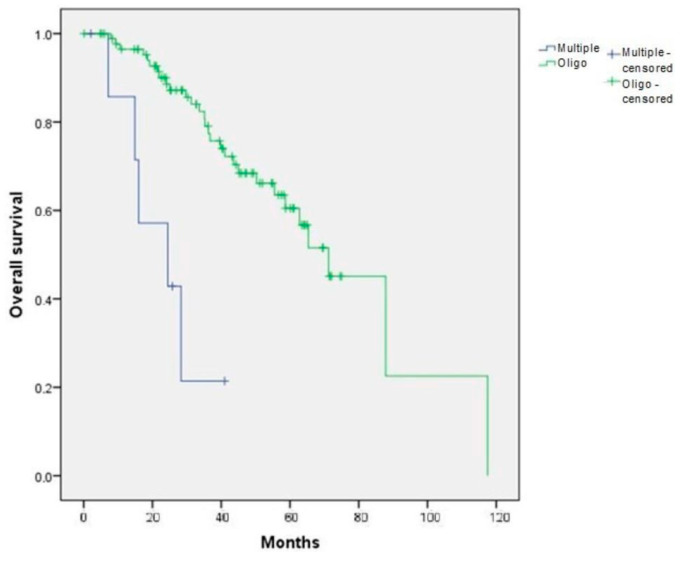
Overall survival depending on the type of first progression (multiple versus oligo) in the initial cohort of 100 patients (OS-LP). *p* < 0.001.

**Figure 3 cancers-15-01195-f003:**
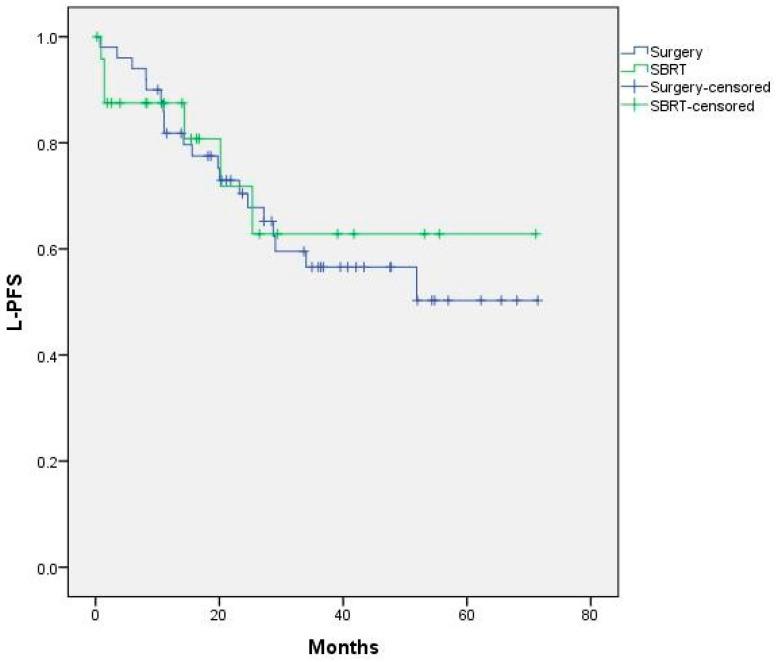
L-PFS curves for surgery vs SBRT. Median not reached. *p* = 0.809.

**Table 1 cancers-15-01195-t001:** Patient and treatment characteristics in LP cohort.

	N = 100
Sex	
Male	69
Female	31
Age at CRC diagnosis (years)	68 (27–85)
>70	40
Initial stage	
I	4
II	27
III	38
IV	30
NS	1
Tumor mutations	
KRAS	
Mut	19
NS	31
WT	15
NRAS	
Mut	3
NS	85
WT	12
BRAF	
NS	83
WT	17
MSI	
No	9
Yes	4
NS	87
Primary tumor surgery	
No	1
Yes	99
Primary tumor location	
Right colon	16
Left colon	35
High rectum	21
Middle rectum	15
Low rectum	13
Neoadjuvant therapy	
Yes	37
No	63
Chemotherapy	
Fluorouracil	13
Capecitabine	3
FOLFOX	6
FOLFOX-Fluorouracil	3
FOLFOX-Anti-EGFR	1
XELOX	1
NS	10
Radiotherapy	
25 Gy	4
50 Gy + boost	14
Other	13
Adjuvant therapy	
No	28
Yes	72
Chemotherapy	
Fluorouracil	3
Capecitabine	10
FOLFOX4	10
FOLFOX6	35
XELOX	10
Other	4

CRC, colorectal cancer; Mut, mutated; WT, wild type; NS, not specified.

**Table 2 cancers-15-01195-t002:** Patient and treatment characteristics in SBRT and surgery groups (FLP cohort).

	Surgery (n = 50)	SBRT (n = 25)	*p*
Sex			0.304
Female	16 (32%)	10 (40%)	-
Male	34 (68%)	15 (60%)	-
Initial stage			0.085
I and II	20 (40.8%)	2 (8%)	-
III	17 (34%)	13 (52%)	-
IV	13 (26%)	9 (36%)	-
NS	0	1 (1%)	-
Chronicity			0.313
Metachronous	46 (92%)	20 (80%)	-
Synchronous	4 (8%)	5 (20%)	-
M1 location			0.00
RLL	11 (22%)	3 (12%)	-
LLL	11 (22%)	5 (20%)	-
Lingula	2 (4%)	0	-
RML	5 (10%)	2 (8%)	-
RUL	12 (24%)	9 (36%)	-
LUL	9 (18%)	6 (24%)	-
Age at M1 diagnosis (years)	75 (29–84)	75 (39–85)	0.184
>70	22 (44%)	14 (56%)	0.606
M1 size (mm)	14.60 (1.1–65)	13.00 (6–40)	0.185
Time to first thoracic progression (months)	27.75 (0–99.38)	27.11 (0–100.34)	0.042

SBRT, stereotactic body radiotherapy; RLL, right lower lobe; LLL, left lower lobe; RML, right middle lobe; RUL, right upper lobe; LUL, left upper lobe.

## Data Availability

The data can be shared upon request.

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
