# Peer review of "Stereotactic Body Radiotherapy versus Surgery for Lung Metastases from Colorectal Cancer: Single-Institution Results"

_cancers, 2023, doi:10.3390/cancers15041195_

Round 1
Reviewer 1 Report
Dear Editor and Authors,
Thank you for asking me to review this quite interesting study titled “Stereotactic body radiotherapy versus surgery for lung metastases from colorectal cancer: single-institution results” By Dr. Garcia-Exposito and colleagues from the Radiation Oncology Department at the Institut Català d’Oncologia in Barcelona, Spain.
In this single institution, retrospective study the authors are analyzing and comparing the outcomes between two groups of patients with metastatic to the lung colorectal cancer, one group comprised of 50 patients who underwent surgical metastasectomy and a second group of 25 patients which underwent stereotactic body radiotherapy (SBRT). They hypothesized that both management approaches would have similar control rates and outcome.
As the authors concede surgery is and remains the treatment of choice! It is not however inconceivable that a work originating from a radiation oncology department would advocate radiotherapy. It is not unexpected as well that no thoracic surgeon is included amongst the authors of this work to provide a different perspective. In addition, this exclusion of surgeons from this work has limited the information available about the surgical group which weakens as one might expect it value (for example the authors note “Surgical treatment was based on the clinical judgment of the surgical team and de-104 tails were not recorded in the present study - lines 104-105”). Therefore, I feel certain issues need to be highlighted so that no one-sided information and messages reach the reader.
To begin the work is not methodologically sound and suffers from significant limitations. Specifically it is retrospective in nature which of course means that specific data points/variables needed are not available or if available there may be a lack of completeness of data. The authors do not mention if they performed a retrospective chart review (which is notorious for producing inconsistent results) or utilized a departmental database which although better in terms of consistency has fixed variables and is limited in terms of the comparisons that can be made and the inclusion of contributing factors.
In addition, the sample size is very, very small. We are talking about 75 patients in total studied of which there is an imbalance and only 25 belong to the SBRT group. The authors have not performed any type of sample size calculation or power analysis to prove that these numbers are enough to demonstrate significant statistical results!! I feel the lack of difference between the two groups in terms of outcome is because there just isn’t enough patients to demonstrate it!!
One note as well regarding ethical issues. Although the authors report that informed consent was obtained of all patients (how was this performed if the study was retrospective and considering a number of patients had died in the meanwhile - overall survival is a secondary end point recorded - is unknown), no ethical approval by the hospital’s investigation board is mentioned. Was one not obtained?
The SBRT treatment was performed as an outpatient, how many sessions on average did the patients have to undergo?
In terms of the statistical analysis, there are a number of issues – as previously mentioned the number of patients (and variables) are limited and thus a multivariable analysis is not possible or very limited in power!!
Given that all patients developed recurrence and metastasis why only 19 received both local and systemic treatment (I suspect chemotherapy or immunotherapy). What percentage of each systemic treatment was utilized (chemo or immune).
Why was one patient not undergone surgery for his primary site?
The tables (especially table 2) need to report the statistical significant results and if possible p-values.
The conclusion the authors report and advocate in their discussion is In the present study, the overall survival at 2 years—90% in the surgery group versus 277 89% in the SBRT group—was superior to a previous series
In addition there are a number of issues that the authors need to mention in their discussion to provide a balanced overview of the two techniques:
- They need to mention that in metastasis having new histology is quite beneficial for the oncological management of the patient. Often delayed (or even early)metastatic disease might have a different molecular and histopathological profile to the primary one. Therefore, having tissue to perform analysis of EGFR, KRAS, ALK, PDL1 ect is quite useful to the oncologist which can tailor their systemic therapy accordingly and utilizing the updated profile of the tumour(s).
- Albeit metastasectomy is a surgical procedure and does have a risk profile and maybe is more “taxing” to the patient it is a definitive treatment with a short duration. For example nowadays with minimally invasive surgery – thoracoscopy/VATS the risk to the patient is minimal, pain is quite reduced, length of stay is 3-4 days and the patient in one session/one shot undergoes definitive and final treatment!!
- A comparison of cost between the two techniques also needs to be provided in the discussion section. It is not certain that SBRT is a cheaper approach considering initial cost for equipment purchase and setup, multiple visits required, ect. I think this is something needing addressing.
In conclusion, although this study really has a lot of deficits and does not merit publication I would like to discuss it with the authors and see if it can be improved, especially if modifying the “absolute” message it is trying to advocate. Therefore, I would like to see the comments made addressed before rendering final judgment. Thank you again for giving me the opportunity to review this work and I wish well to all.
Author Response
Point by point response to reviewers. First of all, we would like to thank you for the opportunity to improve the manuscript with your suggesions/questions. Reviewer1 As the authors concede surgery is and remains the treatment of choice! It is not however inconceivable that a work originating from a radiation oncology department would advocate radiotherapy. It is not unexpected as well that no thoracic surgeon is included amongst the authors of this work to provide a different perspective. In addition, this exclusion of surgeons from this work has limited the information available about the surgical group which weakens as one might expect it value (for example the authors note “Surgical treatment was based on the clinical judgment of the surgical team and de-104 tails were not recorded in the present study - lines 104-105”). Therefore, I feel certain issues need to be highlighted so that no one-sided information and messages reach the reader. Thank you for your point. We carry out this work with the Department of thoracic surgery. The second author, Dr Ramos, is a surgeon with several years of experience. Moreover, the patients who underwent SBRT instead of surgery were presented in a multidisciplinary tumor board (line 78) Regarding your question about surgical details line 104, our primary end point was local thoracic control rates so details such us type of surgery or complications were not recorded. In order to improve the manuscript we have added a sentence in limitations of the manuscript line 301. To begin the work is not methodologically sound and suffers from significant limitations. Specifically it is retrospective in nature which of course means that specific data points/variables needed are not available or if available there may be a lack of completeness of data. The authors do not mention if they performed a retrospective chart review (which is notorious for producing inconsistent results) or utilized a departmental database which although better in terms of consistency has fixed variables and is limited in terms of the comparisons that can be made and the inclusion of contributing factors. Thank you for sharing your concern. In order to clarify your point we have added the sentence :”from de multidisciplinary lung tumor board data base”, at line 65. We want to point out that this review is real life review including all patients presented in the multidisciplinary tumor board with lung metastases from colorectal cancer suitables for local treatment. That data base was designed after a consensus with medical oncologists, surgeons, pneumologists and radiation oncologists. In addition, the sample size is very, very small. We are talking about 75 patients in total studied of which there is an imbalance and only 25 belong to the SBRT group. The authors have not performed any type of sample size calculation or power analysis to prove that these numbers are enough to demonstrate significant statistical results!! I feel the lack of difference between the two groups in terms of outcome is because there just isn’t enough patients to demonstrate it!! Thank you for sharing your thoughts. We are aware about the limitation of our sample size (line 298), however we must keep in mind that this is a retrospective data base review, so the calculations for sample size in a retrospective review is not mandatory. Moreover, there are several manuscripts already published with small sample size in this sense: (Included in references) 23. Filippi AR, Guerrera F, Badellino S, et al. Exploratory Analysis on Overall Survival after Either Surgery or Stereotactic Radiotherapy for Lung Oligometastases from Colorectal Cancer. Clin Oncol (R Coll Radiol). 2016;28(8):505-512. doi:10.1016/j.clon.2016.02.001. surgery (n = 142) and SBRT (n = 28) Lodeweges JE, Klinkenberg TJ, Ubbels JF, Groen HJM, Langendijk JA, Widder J. Long-term Outcome of Surgery or Stereotactic Radiotherapy for Lung Oligometastases. J Thorac Oncol. 2017;12(9):1442-1445. doi:10.1016/j.jtho.2017.05.015 surgery (n = 68) and SABR (n =42). Lee YH, Kang KM, Choi HS, et al. Comparison of stereotactic body radiotherapy versus metastasectomy outcomes in patients with pulmonary metastases. Thorac Cancer. 2018;9(12):1671-1679. doi:10.1111/1759-7714.12880 SBRT n=21; Surgery n=30. One note as well regarding ethical issues. Although the authors report that informed consent was obtained of all patients (how was this performed if the study was retrospective and considering a number of patients had died in the meanwhile - overall survival is a secondary end point recorded - is unknown), no ethical approval by the hospital’s investigation board is mentioned. Was one not obtained? Thank you for giving us the opportunity to answer this point. All patients signed written informed consent prior to treatment. Moreover, the study was presented to the Institutional Review Board/Research Ethics Board committee and it was approved. To clarify this point we have added the following sentence: The patients signed written informed consent prior to treatment (either surgery or radiotherapy). Moreover, the retrospective study was presented to the Institutional Review Board/Research Ethics Board committee and it was approved. Line 76-77 The SBRT treatment was performed as an outpatient, how many sessions on average did the patients have to undergo? Thank you for your question. The SBRT characteristics have been described in lines 179-182, we have not included the average of sessions on the text because the fractionation depends on the location of the lesion. One of the most important factors described to predict local control is the biological effective dose (BED) and the median of BED has been described in line 181. In terms of the statistical analysis, there are a number of issues – as previously mentioned the number of patients (and variables) are limited and thus a multivariable analysis is not possible or very limited in power!! Thank you for this point. We made a univariate analysis and we didn´t find any difference between surgery and SBRT so we don´t performed the multivariate analysis. Corrected in line 149 and Line 209 Given that all patients developed recurrence and metastasis why only 19 received both local and systemic treatment (I suspect chemotherapy or immunotherapy). What percentage of each systemic treatment was utilized (chemo or immune). Thank you for giving us the possibility to clarify this point. From the initial cohort (100 patients), 91 presented Oligo progressive disease (lung and elsewhere) and in this cohort of patients 75 patients had oligoprogressive lung disease. (lines 161-172). From this 75 patients 19 received local and systemic treatment. In table 2, we show the characteristics of these patients, and only a small proportion of patients had synchronous disease (Surgery 4 patients , SBRT 5 patients), to justify local and systemic treatment. On the other hand, the median of time to first thoracic progression was 27.75 and 27.11 for Surgery and SBRT respectively so patients with only one lesion with a large disease free survival the medical oncologists, prioritize local treatments. Regarding the % of each systemic treatment after progression, we agree with the reviewer that it could be interesting for OS and maybe LC. However, the data base review started in 2004, immunotherapy treatments at this point were under investigation. So, in order to not increase the variability we didn’t include these data. Why was one patient not undergone surgery for his primary site? The patient finally underwent surgery for the primary site, however it was deferred due to toxicity to the chemotherapy. The tables (especially table 2) need to report the statistical significant results and if possible p-values. Thank you. We have added these data. The conclusion the authors report and advocate in their discussion is In the present study, the overall survival at 2 years—90% in the surgery group versus 277 89% in the SBRT group—was superior to a previous series In addition there are a number of issues that the authors need to mention in their discussion to provide a balanced overview of the two techniques: 1. They need to mention that in metastasis having new histology is quite beneficial for the oncological management of the patient. Often delayed (or even early)metastatic disease might have a different molecular and histopathological profile to the primary one. Therefore, having tissue to perform analysis of EGFR, KRAS, ALK, PDL1 ect is quite useful to the oncologist which can tailor their systemic therapy accordingly and utilizing the updated profile of the tumour(s). We agree with the review about the benefit of pathological sample, however all these patients , as we mentioned above, were presented to the lung tumor board and in these cases where histological sample were needed we prioritize biopsy or surgery. Lines 294-297. 2. Albeit metastasectomy is a surgical procedure and does have a risk profile and maybe is more “taxing” to the patient it is a definitive treatment with a short duration. For example nowadays with minimally invasive surgery – thoracoscopy/VATS the risk to the patient is minimal, pain is quite reduced, length of stay is 3-4 days and the patient in one session/one shot undergoes definitive and final treatment!! Thank you for your statement. The primary end point of this retrospective data review is to compare local thoracic control. With this exploratory data results from a radiation oncologists and surgeons offering a similar local control, we are planning to start a prospective data analysis including specific surgical characteristics (VATS, robotic-assisted thoracoscopic surgery RATS) and including molecular profile (NGS, ctDNA). 3. A comparison of cost between the two techniques also needs to be provided in the discussion section. It is not certain that SBRT is a cheaper approach considering initial cost for equipment purchase and setup, multiple visits required, ect. I think this is something needing addressing. Thank you for your concern about this interesting point.. We believe it is more interesting to talk about Quality-adjusted Life Year (QUALY) and total Cost-effectiveness. Regarding this issue , the literature review show that SBRT is more cost-effective than VATS in stage I NSCLC (see references below), however in this metastatic populations with heavily treated patients it could be interesting to see if QUALYs and cost-effectiveness would suport that a non invasive radical treatment would be less cost-effective than a minimally invasive treatment (VATS, RATS) In this retrospective review, we did not register these data , however with this exploratory analysis of our tumour board data base , we want to include this data in a prospective cohort to suport our findings. Bibliography: 1. Wolff HB, Alberts L, van der Linden N, Bongers ML, Verstegen NE, Lagerwaard FJ, et al. Cost-effectiveness of stereotactic body radiation therapy versus video assisted thoracic surgery in medically operable stage I non-small cell lung cancer: A modeling study. Lung Cancer Amst Neth. 2020 Mar;141:89–96. 2. Paix A, Noel G, Falcoz PE, Levy P. Cost-effectiveness analysis of stereotactic body radiotherapy and surgery for medically operable early stage non small cell lung cancer. Radiother Oncol. 2018 Sep;128(3):534–40. 3. Sun H, Jin C, Wang H, Hu S, Chen Y, Wang H. Cost-effectiveness of stereotactic body radiotherapy in the treatment of non-small-cell lung cancer (NSCLC): a systematic review. Expert Rev Pharmacoecon Outcomes Res. 2022 Jul;22(5):723–34. 4. Leaman-Alcibar O, Cigarral C, Déniz C, Romero-Palomar I, Navarro-Martin A. Quality of Life After Stereotactic Body Radiation therapy Versus Video-Assisted Thoracic Surgery in Early stage Non-small Cell Lung Cancer. Is there Enough Data to Make a Recommendation?. J Clin Transl Res. 2021;7(2):209-220. Published 2021 Apr 22.
Reviewer 2 Report
Dear authors,
thanks for having the oppurtunity to review your paper.
I have some comments and hope you can address them:
Simple Summary: -
Abstract:
"who during follow-up .."
->Did you include only patients with metachronous metastasis or also patients with synchronous metastasis?
The tables list 30 patients with stage IV UICC. If you include patients with synchronous metastases, SBRT or surgery could be part of the primary treatment. Therefore, please use a term other than "during follow-up".
Introduction:
"Due to the difficulty of designing a prospective trial to compare the three techniques, most knowledge is obtained from retrospective studies"
-> you might better add "and due to the absent of prospective trials ..."
You can add in the introduction section that SBRT is limited for hilar or ultra-hilar metastases because of the toxicity risk.
Materials and Methods
Patients
Reading this part without considering the rest of the manuscript, I have the impression that all patients developed metastasis (metachronous), but if I understood your paper correctly, patients with synchronous metastasis were also included. I would recommend to describe the cohort in more detail. (For example, specify the number of patients of synchronous or metachronous metastasis).
Please add a short consort diagram which shows the LP and FLP cohorts more comprehensible, thanks.
"using an end-expiratory breath-hold technique. "
-> In the next sentence you wrote that you used 4D CT Scan to complete respiratory cylces, too. When we performe stereotactic radiotherapy for lung metastases on a linac we use a inspiration breath holding technique, too but do not describe dose to a ITV as the radiation is always performed in the same breath phase.
Perhaps you can correct this part or explain why I'am wrong, thanks :).
"Surgical treatment"
->missing information about surgical complication/toxicities should be mentioned in the limitation section
"blood test were performed"
->You may CEA test, correct?
"an a/b of 10"
->You may use alpha/beta symbols
Statistics
You say you used several test but in your result section you did not report results from these ! e.g. multivariate cox regression If you do not use a method in your work you should mentioned these in the Statistic section.
Results
see above please add a consort diagramm or another kind of graphical description of the study cohorts
Figure 1
please add number at risk below the figure, please include P-value of log-rank test in all figures if approbiate
Figures can be modifed within SPSS or exported as file for further modification in Powerpoit or adobe pro
-> you should use a "." rather "," in you diagram axes (e.g 0.4 not 0,4)
"Median FollowUp was 36 months"
->Do you use the reverse kaplan maier approch to calculate ?, please add the method you use.
"univariate analysis"
-> where are these results ? please add a supplementary table with these results.
Figure 2
-> see comments for Figure 1
-> as you see, the survival lines are crossing - that may indicate a violation of proportional hazard assumption. It is complicated to test the proportional hazard assumption in SPSS, in R it is easier (http://www.sthda.com/english/wiki/cox-model-assumptions) If you want to provide a p-value you should comment why you do not test for the proportional hazard assumption or it may be easier here just to provide the L-PFS rate at different time points which are similiar and indicate no different in outcomes between both treatments.
Figure A1.1, 1.2,
-> see comments for Figure 1
"p=0.344"
-> what test you used here ?
Figure A3
-> see comments for Figure 1
Figure A4
-> see comments for Figure 1
as mentioned above, in all figure where survival lines cross - test for proportional hazard assumptions should be performed if you report a p-value
Discussion
"additional difficulties when designing a prospective study"
-> and lead to systemic bias when comparing both salvage modalities, plesse add this (SBRT normal older, more comorbities not suitable for surgery ..)
" stratified analysis was performed"
-> please correct me if I'am wrong but where to you performed a stratified analysis?
"In the present study we observed that 100% of the deaths in the surgery cohort were from cancer progression compared with 33% in the SBRT group, indicating than even with the same OS, SBRT had fewer cancer-related deaths."
-> This is a complicated statement - a surgeon could argue that SBRT is so toxic to the lungs and heart that our patients are more likely to die from respiratory failure or heart failure than from tumor growth.
Please comment on this and add another possible explanation for the low number of cancer related death behind the abscopel effect.
"clarifying that long-term data suggest an advantage towards surgery "
-> instead of the term clarifying you may write: but surgery seem to provide better long-term survival
Limitation
- plese add more limitation
-> limitated information on surgical toxicity, limitaed information on systemic treatments, limitation on cause of death
""Local failure after stereotactic body radiation therapy or wedge resection for colorectal pulmonary metastases""
-> This is a paper of Nelson et al. You may include and discuss this in your discussion section because here a surgeon report lower local control rate safter SBRT. (I would say the RT dose seem to be insufficient").
Conclusion
-" for the treatment of lung metastase"
-> please add in colorectal cancer
Author Response
Point by point response to reviewers. First of all, we would like to thank you for your valuable comments to improve the manuscript , we appreciate your point of view and suggestions. Reviewer 2 Abstract: "who during follow-up .."->Did you include only patients with metachronous metastasis or also patients with synchronous metastasis?The tables list 30 patients with stage IV UICC. If you include patients with synchronous metastases, SBRT or surgery could be part of the primary treatment. Therefore, please use a term other than "during follow-up". We agree wito your comment. We have modified and erased “during follow-up” Introduction: "Due to the difficulty of designing a prospective trial to compare the three techniques, most knowledge is obtained from retrospective studies"-> you might better add "and due to the absent of prospective trials ..." Thank you for your suggestion, we have modified the sentence following your comment. You can add in the introduction section that SBRT is limited for hilar or ultra-hilar metastases because of the toxicity risk. Thank you for your comment. We agree with your comment. We have added the following sentence line 56:”Even though the toxicity profile is low, the indication of SBRT in ultracentral lesions (hilar o less than 2cm from airways) is limited due to the risk of high grade of toxicity” Materials and Methods Patients Reading this part without considering the rest of the manuscript, I have the impression that all patients developed metastasis (metachronous), but if I understood your paper correctly, patients with synchronous metastasis were also included. I would recommend to describe the cohort in more detail. (For example, specify the number of patients of synchronous or metachronous metastasis). Thank you. These data were available in table 2. However, we have added the data, line 170. Please add a short consort diagram which shows the LP and FLP cohorts more comprehensible, thanks. Thank you for your comment. We have added Fig 1. "using an end-expiratory breath-hold technique. " -> In the next sentence you wrote that you used 4D CT Scan to complete respiratory cylces, too. When we performe stereotactic radiotherapy for lung metastases on a linac we use a inspiration breath holding technique, too but do not describe dose to a ITV as the radiation is always performed in the same breath phase.Perhaps you can correct this part or explain why I'am wrong, thanks :). Thank you for highlight this mistake. You are rigth,we don´t treat these patients using breath hold technique. During the scan we are using an acoustic signal to notify patients when to breath in and out in order to make the respiratory pattern stable. Patients were treated using this pattern previosly recorded during the simulation day. We have added a correct sentence line 125. "Surgical treatment" ->missing information about surgical complication/toxicities should be mentioned in the limitation section Thank you we have added a sentence line 340. "blood test were performed" ->You may CEA test, correct? Thank you . You are correct., we have added a sentence line 149. "an a/b of 10" ->You may use alpha/beta símbols Thank you. Changed line 166. Statistics You say you used several test but in your result section you did not report results from these ! e.g. multivariate cox regression If you do not use a method in your work you should mentioned these in the Statistic section. Thank you, we have modified and erased multivariate. Line 187. Results see above please add a consort diagramm or another kind of graphical description of the study cohorts Figure 2 please add number at risk below the figure, please include P-value of log-rank test in all figures if approbiate Thank you for your comment. We have added the p-value in all figures. Figures can be modifed within SPSS or exported as file for further modification in Powerpoit or adobe pro -> you should use a "." rather "," in you diagram axes (e.g 0.4 not 0,4) Thank you we have modified this. "Median FollowUp was 36 months" ->Do you use the reverse kaplan maier approch to calculate ?, please add the method you use. We used SPSS for this calculation. Line 193 "univariate analysis" -> where are these results ? please add a supplementary table with these results. We are not sure about what are you asking for. We have several figures with these variables such as A1.1 with p values. Figure 2 -> see comments for Figure 1 Thank you. -> as you see, the survival lines are crossing - that may indicate a violation of proportional hazard assumption. It is complicated to test the proportional hazard assumption in SPSS, in R it is easier (http://www.sthda.com/english/wiki/cox-model-assumptions) If you want to provide a p-value you should comment why you do not test for the proportional hazard assumption or it may be easier here just to provide the L-PFS rate at different time points which are similiar and indicate no different in outcomes between both treatments. Thank you for your comment and kind explanation. We don´t know how to do that in R. We have described a L-PFS rate at different time points. Line 249-250. Figure A1.1, 1.2, -> see comments for Figure Done "p=0.344" -> what test you used here ? chi squared test, added in line 219 Figure A3 -> see comments for Figure 1 Done Figure A4 -> see comments for Figure 1 Done as mentioned above, in all figure where survival lines cross - test for proportional hazard assumptions should be performed if you report a p-value Discussion "additional difficulties when designing a prospective study" -> and lead to systemic bias when comparing both salvage modalities, plesse add this (SBRT normal older, more comorbities not suitable for surgery ..) Thank you. " stratified analysis was performed" -> please correct me if I'am wrong but where to you performed a stratified analysis? Done table We have added a column in table 2. "In the present study we observed that 100% of the deaths in the surgery cohort were from cancer progression compared with 33% in the SBRT group, indicating than even with the same OS, SBRT had fewer cancer-related deaths." -> This is a complicated statement - a surgeon could argue that SBRT is so toxic to the lungs and heart that our patients are more likely to die from respiratory failure or heart failure than from tumor growth. Please comment on this and add another possible explanation for the low number of cancer related death behind the abscopel effect. Thank you for sharing your thougths , We agree that we have to be cautious, with this statment. We report no GIII toxicity, in the SBRT cohort. Moreover, We want to highlight that even that SBRT population is more fragile than surgical population the OS is the same. We have added this sentence. :” that population treated with SBRT is more fragile but also supporting the abscopal theory.” Line 308. "clarifying that long-term data suggest an advantage towards surgery " -> instead of the term clarifying you may write: but surgery seem to provide better long-term survival Done Line 331 Limitation - plese add more limitation -> limitated information on surgical toxicity, limitaed information on systemic treatments, limitation on cause of death Done ""Local failure after stereotactic body radiation therapy or wedge resection for colorectal pulmonary metastases"" -> This is a paper of Nelson et al. You may include and discuss this in your discussion section because here a surgeon report lower local control rate safter SBRT. (I would say the RT dose seem to be insufficient"). Thak you for sharing this. We have added this paper. Line 324. Conclusion -" for the treatment of lung metastase" -> please add in colorectal cancer Done
Round 2
Reviewer 1 Report
Dear Editor and Authors,
So I have re-read and re-evaluated your revised manuscript. I do remain unconvinced by your hypothesis and I question that there is no bias (personal and scientific (, included in this analysis!!
On the other hand, it is evident you have put a lot of work onto this, you have made the corrections asked to the extend possible and have acknowledged the limitations of the work. Therefore, even though I still consider it controversial and unproven it is not ethical for me as a reviewer to reject your work because I don’t believe in SABRT (that would be interjecting my own bias as a thoracic surgeon and would not be good and ethical science!!). It is thus up to the thoracic surgical community to rebuttal your claims with scientific arguments and new research in the future.
Therefore, I am recommending its acceptance for publication to generate interest and discussion and promote further research on the matter. Good luck.
Kind regards to all,
Reviewer 2 Report
Dear authors,
thanks for taking the time to address my comments.
I have no further comments.